# The Contribution of Citizens to Community-Based Medical Education in Japan: A Systematic Review

**DOI:** 10.3390/ijerph18041575

**Published:** 2021-02-07

**Authors:** Ryuichi Ohta, Yoshinori Ryu, Chiaki Sano

**Affiliations:** 1Community Care, Unnan City Hospital, 96-1 Iida, Daito-cho, Unnan 699-1221, Shimane, Japan; yoshiyoshiryuryu.hpydys@gmail.com; 2Department of Community Medicine Management, Faculty of Medicine, Shimane University, Izumo 693-8501, Shimane, Japan; sanochi@med.shimane-u.ac.jp

**Keywords:** community-based medical education, Japanese medical education, comprehensive medical training

## Abstract

Community-based medical education (CBME) offers vital support to healthcare professionals in aging societies, which need medical trainees who understand comprehensive care. In teaching comprehensive care practices, CBME can involve citizens from the relevant community. This research synthesizes the impact of the involvement of communities on the learning of medical trainees in CBME. We conducted a systematic review, in which we searched ten databases from April 1990 to August 2020 for original articles in Japan regarding CBME involving citizens and descriptively analyzed them. The Kirkpatrick model was used to categorize the outcomes. Our search for studies following the protocol returned 1240 results; 21 articles were included in this systematic review. Medical trainees reported satisfaction with the content, teaching processes, and teachers’ qualities. Medical trainees’ attitudes toward community and rural medicine improved; they were motivated to become family physicians and work in communities and remote areas. This review clarified that citizen involvement in CBME had an effective impact on medical trainees, positively affecting perceptions of this type of education, as well as improving trainees’ knowledge about and attitude toward community and rural medicine.

## 1. Introduction

Community-based medical education (CBME) is a practical educational method in which medical trainees learn primary healthcare and primary care in medical institutions outside of medical universities and tertiary hospitals [1,2,3]. CBME is vital for healthcare professionals to learn about aging societies. Through real experiences with CBME, medical students and residents can learn practical knowledge, expertise, and attitudes regarding clinical reasoning, interprofessional collaboration, and community medicine, which they cannot learn in medical universities [4,5,6,7]. Aging societies need medical trainees who understand comprehensive care [8,9,10,11]. Comprehensive care refers to an approach to care in which communities are able to address citizens’ healthcare comprehensively and in collaboration with specialized medics from outside those communities [12]. Older people lose access to medical care because of their loss of mobility, especially in rural areas where public transportation is minimal [11]. Comprehensive care can be important in enabling older people to access healthcare and maintain their health [13,14]. Furthermore, for effective, comprehensive care, medical trainees should be educated about comprehensive care, including the perspectives of citizens in communities.

To teach comprehensive care to medical trainees, CBME can actively involve citizens within communities. In administering comprehensive care, medical trainees have to deal with the various problems experienced by older people [15,16]. Elderly patients tend to experience numerous health problems leading to frailty, and these health problems need to be managed through multidisciplinary collaboration in the community [17,18,19]. Citizens in communities can modify the process of managing these health problems, and this can be learned through participation in such communities [20,21]. As citizen involvement in education, CBME, including active participation in community activities driven by citizens, is highly relevant in Japan, the world’s fastest-aging society [22,23,24]. Various educational approaches have emerged from the Japanese context offering medical trainees clinical experience to absorb clinical knowledge, apply it, and understand the community and rural medicine [23,24]. Japanese CBME includes various communities to facilitate medical trainees’ education regarding comprehensive care from hospitals to clinics and urban to rural settings [24]. CBME involving citizens’ activities can be beneficial for the training of medical students and residents in comprehensive care to support the super-aging population in Japanese society. As Japan has led among aging societies all over the world, with the rate of people over 65 years old exceeding 26% in 2015 [20], education in such communities can be specific, and this clarification of the present conditions is beneficial for informing and optimizing education programs in all countries that are preparing for aging populations.

Since the application of CBME in Japan, medical institutions have provided CBME in numerous contexts, which could constitute to evidence of different outcome levels, based on the Kirkpatrick model regarding medical trainees’ perceptions of patient outcomes [25]. Educational research can clarify the effectiveness of CBME and its challenges in Japan in an authentic way. Different contexts of CBME could produce different learning contexts for and perceptions of medical students because of the variation in communities [26,27]. The comprehensive effects of citizens’ involvement in CBME have not yet been clarified by scientific research. A systematic review of scientific evidence in the context of Japan can elucidate the effectiveness of citizens’ involvement in CBME for medical trainees and their concrete learning contents and perceptions. Thus, our research question is as follows:

“What effect(s) does citizens’ involvement in CBME have on medical students and residents?”

To date, no systematic review has focused on the effects of citizens’ involvement in CBME. The purpose of this research is to systematically review the impact of the involvement of communities on the learning of medical trainees in CBME.

## 2. Materials and Methods

### 2.1. Study Design

Systematic review.

### 2.2. Search Strategy

This study followed guidelines stipulated in the preferred reporting items for systematic reviews and meta-analyses (PRISMA) statement. We searched ten databases (PubMed, the Cochrane Library, Google Scholar, CINAHL, Web of Science, Scopus, Embase, Ichushi-Web, Jdrea-mIII, and CiNii) for original articles from Japan regarding CBME from April 1990 to August 2020. Regarding articles written in English, our search strategy was based on the following title/abstract keywords: (“community-based medical education” or “community-oriented medical education” or “CBME” or “COME”) AND (“Japan”). We searched Japanese articles based on the following title/abstract keywords: (“Chiiki-kibangata-igakukyouiku” or “Chiiki-shikousei-igakukyouiku” or “Chiiki-sankagata-igakukyouiku” or “Chiiki-iryozishyu”) and (“Japan”). The reference lists of relevant studies were also reviewed to identify research that might have been missed in the database search.

Ichushi-Web is an online Japanese literature search system provided by the non-profit Japan Medical Abstracts Society [28]. The Ichushi-Web database covers about 10 million medical papers from 6000 journals in Japan and is often used to search Japanese literature (Ichushi-Web). JDreamIII (Japan Science and Technology Agency Document Retrieval System for Academic and Medical Fields) is an online Japanese database for searching literature provided by the Japan Science and Technology Agency [29]. The database of JDreamIII contains roughly 60 million articles, including serial publications, reports, conference materials, public documents, and proceedings on science and technology (JDreamIII). CiNii is an online Japanese literature search system provided by the National Institute of Informatics [30]. The CiNii database contains about 18 million articles focusing on natural and cultural sciences (CiNii books). 

### 2.3. Inclusion and Exclusion Criteria

Literature searches and data extraction were independently conducted by two investigators (RO and YR), and any discrepancies were resolved through discussion. In this study, databases were searched for empirical studies on CBME in Japan to evaluate the impact of CBME involving citizens on medical students and residents. Studies conducted without any clear description of aim, participants, or outcomes were excluded. Details of the inclusion criteria are shown in Table 1.

### 2.4. Data Extraction

One of the investigators (RO) extracted data from each original article using a purpose-designed data extraction form based on the best evidence medical education (BEME) coding form [31,32]. Next, investigators (YR and CS) checked the extracted data. Extracted data were categorized into settings (urban or rural, hospital or clinic, the duration of CBME, the purpose of CBME) and study methodology (qualitative, quantitative, or mixed, data source, outcomes).

### 2.5. Analysis

The quality of each study was assessed based on the BEME scale (1 to 5): Grade 1 indicated that no definite conclusions could be drawn, that is, the data were not significant; Grade 2 indicated that the results were ambiguous, but there appeared to be a trend; Grade 3 indicated that conclusions could probably be drawn based on the results; Grade 4 indicated that the results were clear and very likely to be true; Grade 5 indicated that the results were unequivocal [31,32]. The outcomes of the study were categorized based on the Kirkpatrick outcome evaluation (Level 1, 2A, 2B, 3, 4A, 4B): Level 1 indicated the reaction of the participants to the education; Level 2A indicated changes in learners’ attitudes; Level 2B indicated changes in learners’ knowledge and skills; Level 3 indicated changes in learners’ behaviors; Level 4A indicated changes in the system/organizational practice; Level 4B indicated changes in patient care outcomes [25]. The contents were the results, settings, learning purposes, year of publication, participants, evaluation methods, Kirkpatrick outcome evaluation levels, the quality of studies, and the main outcomes of CBME. As there were various grading systems at different medical schools around the world, participants’ grades were briefly categorized as first year, pre-clinical year, post-clinical year, or final year. As some studies contained participants from multiple categories, we calculated each category independently. For example, if one study contained first-year and pre-clinical-year students, we categorized the study in both categories. To describe the setting, population density (people/square kilometer) was used and classified based on the criteria of the OECD (less than 150 people/square kilometer) [33]. Where there were multiple educational sites across cities and prefectures, the population density was calculated as the average of the population densities involved in the research. The duration of training and training settings were collected for the CBME format. The training settings were categorized as hospitals, clinics, welfare facilities, home care settings, and other (rehabilitation, health promotion, and working experience in other professions). The CBME formats were compared with a nationwide survey on CBME in Japan regarding learning durations and settings [24]. The results were categorized, and the rate of statistically significant results in each category was calculated for each Kirkpatrick outcome evaluation level. When there was no statistical analysis or pre- and post-comparison, the scores above the middle (e.g., three on five-point Likert scales or 50 on the visual analog scale) were considered significant. The Unnan City Hospital Clinical Ethics Committee approved this study (Approval code: 20200020).

## 3. Results

### 3.1. Search Results

In searching for studies that followed the above-mentioned inclusion criteria, 1240 studies were detected. A total of 1188 studies were excluded because they were unrelated to the efficacy of CBME. After reviewing the full texts, 31 studies were excluded for the following reasons: Ten studies did not involve learning via citizens from communities, ten studies did not involve medical students, seven studies did not deal with the outcomes of CBME, and four studies were not original articles (Figure 1). Twenty-one studies were included in this systematic review.

### 3.2. Study Characteristics

Four studies were performed in areas with a population density of fewer than 50 people per square kilometer. Eight studies were performed in areas with a population density of 50 to 150 people per square kilometer. Eight studies were performed in areas with a population density of 150 to 1000 people per square kilometer. One study was performed in an area with a population density of more than 1000 people per square kilometer. Twelve studies were performed in rural areas. Seven studies included first-year medical students, three studies included medical students in their pre-clinical year, ten studies included medical students in their clinical year, seven studies included medical students in their final year, and two studies included medical residents (Table 2).

Eleven of the studies used quantitative methods, eight used qualitative methods, and two used mixed-methods. The quantitative studies’ respective methods comprised four cross-sectional designs, three descriptive designs, and six pre- and post- design studies. The qualitative studies’ respective methods comprised three descriptive analyses, four thematic analyses, two SCATs (Steps for Coding and Theorization), and one ethnography. The concrete data collection methods comprised one direct observation, one e-portfolio, nine interviews, two focus groups, and 11 questionnaires (Table 3).

### 3.3. CBME Format

Two studies were performed over the course of three days, nine studies were performed over one week, eight studies were performed over two weeks, one study was performed over four months, and one study was performed over two years. As the national survey showed that the median duration was one week [24], the studies included in this systematic review had a similar distribution to the national survey. Seventeen studies (80.9%) were set in hospitals, 15 studies (71.4%) were set in clinics, 17 studies (80.9%) were set in welfare facilities, 17 studies (80.9%) focused on home care, and 21 studies (57.6%) focused on community settings involving citizens’ activities (Table 4).

### 3.4. Kirkpatrick Outcome Levels and the Strength of the Findings

Twelve studies were evaluated at Kirkpatrick level 1, fourteen at level 2A, and fifteen at level 2B. No study was assessed with level 3 or 4 Kirkpatrick outcomes. Among the reviewed quantitative studies, seven studies were assessed at level 1, eight studies at level 2A, and six studies at level 2B. Among the reviewed qualitative studies, three studies were assessed at Kirkpatrick level 1, four studies at level 2A, and seven studies at level 2B. Among the mixed-method studies that were reviewed, two studies were assessed at Kirkpatrick level 1, two studies at level 2A, and two studies at level 2B. Regarding the findings of the reviewed studies, one study was categorized as Grade 2, twelve studies as Grade 3, and seven studies as Grade 4 (Table 5).

### 3.5. Synthesis of Findings on the Impact on Stakeholders in CBME

Table 6 describes the details of each study that evaluated the impact of Japanese CBME on medical trainees.

### 3.6. Kirkpatrick Outcome Level 1

#### 3.6.1. Quantitative Assessment: Satisfaction with the Training, Overall Quality of the Training, Quality of the Teachers, Quality of the Contents, and Importance of the Training

The quantitative outcome contents at Kirkpatrick level 1 in this study consisted of five categories: Satisfaction with the training, overall quality of the training, quality of the teacher, quality of the contents, and importance of the training. Regarding respondents’ satisfaction with training, one study returned significantly high results [34]. Regarding the quality of the training, 5/5 of the studies returned significantly high results [34,42,44,46,48,52]. Regarding the quality of the teacher, 5/5 of the studies returned significant results [34,42,44,46,48,52]. Regarding the quality of the content, 3/4 of the studies returned significant results [42,44,48,52]. Regarding the importance of the training, 4/4 of the studies returned significant results [34,42,44,52,53].

#### 3.6.2. Qualitative Assessment: Noteworthy Interaction with Citizens, the Distance between Medical Institutions, and the Motivation of Citizens and Professionals

Medical trainees have difficulty scheduling CBME given its typical distance from main hospitals, as well as a lack of support from main hospitals [46,48]. In addition, medical trainees hoped to train in CBME for a longer period of time [54]. Medical trainees had noteworthy experiences through their interactions with indigenous people and experienced satisfaction [46,48,52]. In contrast, the lack of motivation for education among rural people had a negative impact on medical trainees’ learning [48,54]. In the personal axis category, the establishment of effective relationships with healthcare professionals and citizens motivated respondents to work in rural and remote areas [48,52].

### 3.7. Kirkpatrick Outcome Level 2A

#### 3.7.1. Quantitative Assessment: Interacting with Patients/Citizens, Motivation for General Practitioners, Motivation for Specialists, Motivation for Working in Community Medicine, and Motivation for Working in Remote Areas

The quantitative outcome contents at Kirkpatrick level 2A in this study consisted of five categories: Interacting with patients/citizens, motivation for general practitioners, motivation for specialists, motivation for working in community medicine, and motivation for working in remote areas. All assessments were conducted using a questionnaire. Regarding interacting with patients/citizens, 2/2 of the studies returned significant results [34,52]. Regarding motivation for general practitioners, 5/5 of the studies returned significant results [34,41,46,52,53]. Regarding motivation for specialists, 4/4 of the studies returned significant results [34,41,46,53]. Regarding motivation for working in community medicine, 4/4 of the studies returned significant results [36,46,48]. Regarding motivation for working in remote areas, 4/4 of the studies returned significant results [36,39,41,53].

#### 3.7.2. Qualitative Assessment: The Relationship between Professionals and Citizens, Strong Connections between Citizens for Health, and Motivation for Learning Rooted in the Expectations of the Citizens

Medical trainees observed tension from the competing demands between healthcare professionals and citizens [53], as well as the importance of a strong sense of connection among the islanders [46,50,51]. Through interactions with citizens, medical trainees realized their lack of knowledge and were motivated to learn more by feeling the expectations of the community members [46,51,53,54]. Medical trainees also observed the need for a shift from physician-centered to citizen-centered perspectives through the observation of healthcare professionals’ interaction with citizens in their communities [46,51,53], which generated motivation for general physicians [52,54].

### 3.8. Kirkpatrick Outcome Level 2B

#### 3.8.1. Quantitative Assessment: Community Medicine, Remote Medicine, Citizens’ Lives, and Preventative Medicine

The quantitative outcome contents at Kirkpatrick level 2B in this study consisted of four categories: Community medicine, remote medicine, citizens’ lives, and preventative medicine. All assessments were conducted using a questionnaire. Regarding community medicine, 4/4 of the studies returned significant results [34,36,39,41]. Regarding remote medicine, 2/2 of the studies returned significant results [36,41]. Regarding citizens’ lives, 2/2 of the studies returned significant results [40,52]. Regarding preventable medicine, 2/2 of the studies returned significant results [40,52].

#### 3.8.2. Qualitative Assessment: The Significance of Citizens’ Characteristics, Humanistic Relationships, and Community-Oriented Primary Care

Through their interactions with citizens in communities, medical trainees came to understand the significance of local characteristics, such as culture [46,48,52], as well as the indigenous lives of community members, from children to older people, as related to their health conditions [46,48,52,54]; trainees additionally learned the value of human relationships [45,48,53], expansion of the world through social existence [53], and community-oriented primary care [53].

## 4. Discussion

This systematic review demonstrated the efficacy of and challenges faced by Japanese CBME with citizen participation based on the Kirkpatrick model framework. Based on the quantitative analysis, the present Japanese CBME can be understood to provide an excellent experience and effective learning for medical trainees from Kirkpatrick level 1 to 2B, lacking the assessment of Kirkpatrick level 3 to 4 outcomes due to the scarcity of long-term learning via Japanese CBME. In the qualitative analysis, various content was learned by medical trainees and evaluated at Kirkpatrick levels 1 to 2B. There are various suggestions regarding Japanese CBME systems for better quality regarding the schedule management and motivation of stakeholders involved in CBME.

Japanese CBME can provide satisfactory and practical education for medical trainees over various durations of training. Based on our quantitative analysis, the most common duration of CBME was one to two weeks, which is generally shorter than that in other countries, such as Australia and Canada [7,55]. This review shows that trainee satisfaction and the quality of the contents and teachers in CBME were all highly rated over a comparatively shorter duration of training. The issues of Japanese medical education surrounding university-centered education could have affected this. In Japanese medical education, medical trainees have few opportunities to learn in community-based settings [16,18]. Their learning in universities tends to be based on lectures and observation of their teachers in hospitals [56]. Medical trainees can experience and participate actively in various clinical situations by learning via CBME [6,57]. The educational gap between universities and CBME programs could render their perception of CBME fruitful [58]. In other countries, there is considerable evidence showing the improvement of medical trainees’ educational outcomes in CBME environments compared to medical universities’ education [1,4,7,59]. The present CBME can be promoted more actively to improve medical students’ and residents’ educational environments.

Training in CBME can also motivate trainees to work in communities and remote places [7,60], as shown by this review’s results. In Japan, as CBME has become prevalent, this improvement in the quality of education could serve to drive the allocation of medical doctors in communities, which could function to solve the problem of the localization of physicians in Japan [20,61]. As an effect of CBME, the increased allocation of physicians in rural areas can positively change rural people’s health conditions. The changes can be assessed by systems approaches or multimedia approaches, which can deal with multifactorial interventions in communities not only from medical institutions, but from whole communities as well, including citizens’ activities [62,63]. Future studies can investigate the comprehensive effects of CBME, including citizens’ activities, based on the new methods.

Furthermore, longitudinal integrated clerkship (LIC) can be applied in Japanese medical education, leading to an improvement of the lack of research evaluating Kirkpatrick level 3 and 4 outcomes as LIC is becoming popular in various developed countries to increase medical trainees’ educational outcomes, as well as the number of physicians available in remote areas [3,64,65]. In rural areas, medical teachers and medical professionals are lacking, so the performance of LIC should be strongly supported by local governments and medical universities. For LIC, multiple stakeholders from different institutions should collaborate effectively, which can lead to better education and outcomes regarding medical students’ and residents’ learning. Further research should investigate the LIC application process regarding multisectoral collaboration and the long-term outcomes of CBME, including citizens.

Based on the qualitative analysis, medical trainees can experience biomedical factors, psychosocial factors, and community issues in their interactions with physicians, other medical professionals, and citizens/patients. In addition, they experienced several difficulties, such as differences in educational methods between medical universities and community medical institutions and mentoring systems in medical universities. Furthermore, the low motivation of stakeholders in CBME can affect medical trainees’ motivation negatively. The educational methods may differ between medical universities and CBME because of the different range of diseases dealt with in each setting. In addition, the lack of medical resources in community medical institutions enables medical trainees to participate actively in-patient care, which can benefit medical trainees and medical institutions [66,67]. CBME, supported by a rigid safety system of medical trainees, can drive medical trainees’ learning, more in Japan [68]. In addition, stakeholders should be motivated to provide CBME adequately [69,70,71]. For their motivation, the benefits of CBME for their conditions should be emphasized, such as retention and recruitment of medical professionals, especially in rural areas. In Japanese contexts, there is emerging evidence to demonstrate CBME’s effect on the motivation of medical trainees and the increased number of medical professionals [72,73]. Therefore, subsequent studies should investigate this impact at Kirkpatrick level 3 to 4 outcomes. Currently, the new coronavirus is impinging on community-based medical education because of the risk and fear of spreading infection among trainees, stakeholders, and citizens. To overcome the challenges presented by the new coronavirus, stakeholders should discuss the advantages and disadvantages of CBME, including citizens, and appropriately apply this educational method with medical trainees [74,75,76].

## 5. Limitation

One of the limitations is that all of the quantitative assessments in research regarding CBME in Japan were performed using only questionnaires, not observations, which can inhibit the assessment of the behavior of medical trainees, as well as patient care quality at Kirkpatrick 3 and 4 level outcomes. Future studies should use longitudinal research designs to assess high-level outcomes, which could motivate more medical educators to begin using CBME. Additionally, for assessing CBME’s effects on citizens, patient-reported outcomes, such as quality of life and self-rated health, can be included as Kirkpatrick 3 and 4 level outcomes. Another limitation was the lack of meta-analysis in this systematic review. The lack of meta-analysis is one of the limitations of this systematic review due to the variety of study designs, participants, and outcome settings, and educational methods. The reviewers are Japanese and specialize in medical education, so this may have introduced bias in the analysis and synthesis of our results. Furthermore, owing to accessibility limitations, the review may have missed other studies published in this research area. To overcome this limitation, we employed all of the search engines privileged in Japan and all over the world. As CBME can be affected by context, there are difficulties in applying educational methods in different contexts. As the world is gradually aging, all countries can experience issues of aging societies, which require comprehensive care respecting each local citizen; the results of this study regarding Japanese CBME’s involvement of citizens can provide suggestions regarding the importance of CBME, including citizens for education to enable medical professionals to prepare for aging societies even in countries that are not currently experiencing aging societies. Future reviews can include CBME in other contexts around the world for specific focuses.

## 6. Conclusions

This systematic review clarified that Japanese CBME with citizen participation provided effective education for medical trainees, which can contribute to their effective understanding of comprehensive care and motivation to work in communities. Since the outcomes are limited to transient outcomes, such as the perception of the education, knowledge, and attitude toward community and rural medicine, future studies should examine the longitudinal effect of CBME, such as medical trainees’ behaviors and patients’ outcomes. The involvement of various stakeholders in CBME, as well as better collaboration, should be established to support effective CBME.

## Figures and Tables

**Figure 1 ijerph-18-01575-f001:**
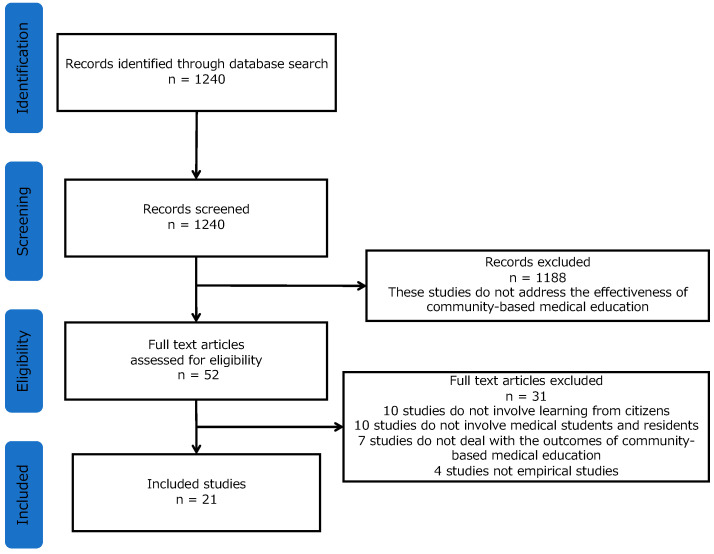
The flowchart of the study population.

**Table 1 ijerph-18-01575-t001:** Inclusion and exclusion criteria.

Criteria	Inclusion	Exclusion
Population	Medical students, residents	Other health care professionals (nurses, pharmacists, dentists, rehabilitators, care managers)
Intervention	Clinical experience focusing on community-based medical education including citizens in communities	Clinical experience only in healthcare facilities
Type of Study	Qualitative, quantitative, mixed-method	Non-empirical studies (editorials, news)
Other	Abstract available Year of publication >1990 Conducted in Japan Including outcome of the participants categorized according to the Kirkpatrick model Full text available in English or Japanese	Abstract not available Full text not available in English or Japanese

**Table 2 ijerph-18-01575-t002:** Distribution of the reviewed studies according to population density and participants’ backgrounds.

Variable		%
**Population Density**		
Fewer than 50	4	19.0%
50 to 150	8	38.1%
150 to 1000	8	38.1%
More than 1000	1	4.8%
Rural (less than 150)	12	57.1%
**Participants**		
First year	7	33.3%
Pre-clinical years	3	14.3%
Clinical year	10	47.6%
Final year	7	33.3%
Medical Resident	2	9.5%

**Table 3 ijerph-18-01575-t003:** Distribution of the reviewed studies according to design and data sources.

Variable	Number of Studies
**Study Method**	
Quantitative method	11
Qualitative method	8
Mixed-method	2
**Study Design**	
**Quantitative**	
Pre- and post-design	6
Descriptive design	3
Cross-sectional	4
**Qualitative**	
Descriptive analysis	3
Thematic analysis	4
SCAT (Steps for Coding and Theorization)	2
Ethnography	1
**Data Resource**	
**Quantitative**	
Questionnaire	11
**Qualitative**	
interview	9
focus group	2
direct observation	1
e-portfolio	1

**Table 4 ijerph-18-01575-t004:** Distribution of the reviewed studies according to community-based medical education (CBME) format.

Variable	Number of Studies
**Duration of CBME**	
Three days	2
One week	9
Two weeks	8
Four months	1
Two years	1
**Study Place**	
Hospital	17
Clinic	15
Welfare facility	17
Home care	17
Community	21

**Table 5 ijerph-18-01575-t005:** Distribution of the reviewed studies according to Kirkpatrick outcome levels and the strength of the findings.

Variable	Number of Studies			
Kirkpatrick Outcome Levels	Quantitative	Qualitative	Mixed	Total
1: Reaction	7	3	2	12
2A: Learning-change in attitudes	8	4	2	14
2B: Learning-change in knowledge and skills	6	7	2	15
3: Change in behaviors	0	0	0	0
4A: Results-change in the system/organizational practice	0	0	0	0
4B: Change in patient care outcomes	0	0	0	0
**Strength of Findings**				
Grade 1: No clear conclusions can be drawn; not practice	0	0	0	0
Grade 2: Results ambiguous, but there appears to be a trend	0	1	0	1
Grade 3: Conclusions can probably be drawn based on the results	6	6	0	12
Grade 4: Results are clear and very likely to be true	4	1	2	7
Grade 5: Results are unequivocal	0	0	0	0

**Table 6 ijerph-18-01575-t006:** Studies reporting the impact of Japanese CBME on medical trainees.

	CBME Setting, Participants/Duration/Population Density	Study’s Purpose	Findings
**Quantitative Method**			
**Okayama et al****.****(2004)** [34]	Clinical and final grades/10 days/ 335 people/km^2^	To examine the effects of a standardized program for medical facilities and clerkship contents introduced in 2001.	The trainees were motivated and had the confidence to talk with citizens in communities regarding medical and social issues.
**Takayashiki et al****.****(2005)** [35]	Final year/10 days/ 335 people/km^2^	To inquire as to the changes in medical students’ perceptions on the necessity of experience in community-based learning programs	The necessity of learning in communities was more likely to be recognized by students who had experienced CBME.
**Tani et al****.****(2009)** [36]	Clinical year/5 days/ 66 people/km^2^	To inquire as to the efficacy of the primary care practice	CBME increased the intensity of students’ interest in and passion for collaborating with citizens in communities as opposed to lectures.
**Okayama et al****.****(2011)** [37]	Clinical year/10 days/ 335 people/km^2^	To clarify which learning activities affect students’ attitudes toward community health care	Health education with citizens was associated with a positive change in both attitudes of “worthiness” (adjusted RR: 1.71, 95% CI: 1.10–2.66) and “confidence” (1.56, 1.08–2.25) for community medicine.
**Okayama et al****.****(2011)** [38]	Clinical year/10 days/ 335 people/km^2^	To explore the association between students’ evaluations of their community- based clinical clerkship, their attitudes toward community health care, and their career preferences	Evaluations of the programs (*p* = 0.014) and students’ attitudes (*p* < 0.001) were strongly associated with an increased preference for a career as a primary care physician after the clinical clerkship.
**Hashiba et al****.****(2011)** [39]	First and pre-clinical years/3 days/ 129 people/km^2^	To assess the impact of the student-led program on students’ notions about, appreciation of, and attitudes toward community- based medicine	The participants showed moderate-to-marked willingness to work in rural areas after their experiences interacting with citizens in communities.
**Iwasaki et al****.****(2011)** [40]	Clinical and final years/5 days/ 108 people/km^2^	To examine the changes in students’ thinking about an affinity for community medicine	Community-based medical programs enhance medical students’ understanding of and affinity for community medicine.
**Tani et al****.****(2014)** [41]	Clinical and final years/5 days/ 66 people/km^2^	To evaluate the effect of community-based clinical education on students’ attitudes toward community medicine and medicine in remote areas.	The intensity of students’ interest and their senses of fulfillment and passion for medicine in remote areas were significantly increased after CBME.
**Katsube et al****.****(2016)** [42]	Clinical and final years/10 days/ 64.4 people/km^2^	To clarify the learning and revising points regarding CBME in rural community hospitals	Based on the questionnaires, the medical students were satisfied with CBME in rural community hospitals.
**Tani et al****.****(2017)** [43]	Clinical year/5 days/ 66 people/km^2^	To examine the effect of community-based clinical practice on their attitudes toward remote medicine and their course after graduation.	Students demonstrated a significantly decreased desire to become general practitioners compared to becoming specialists; this was seen in the students that had a low intensity sense of fulfillment.
**Moriwaki et al****.****(2018)** [44]	Clinical and final years/10 days/ 64.4 people/km^2^	To investigate the change in medical students’ motivation through community-based medical education by surgeons	The medical students’ perceptions regarding rural medicine changed positively regarding the importance of rural medicine.
**Qualitative Method**			
**Yamada et al. (2010)** [45]	First year/5 days/ 26 people/km^2^	To inquire what medical students have actually learned from their involvement in communities	Medical students came to understand life on a remote island through their interactions with citizens.
**Nakada et al. (2010)** [46]	First year/5 days/ 26 people/km^2^	To clarify what students learned on a rural island	Through CBME on a rural island, medical students felt and learned how closely nature and people relate in addition to learning about life on a remote island and the health conditions of the people on the island and recognizing the lack of study and self-realization.
**Uehara et al. (2011)** [47]	First and pre-clinical years/5 days 184 people/km^2^	To discover what medical students learn and accomplish in CBME in rural Japan	The CBME not only can increase students’ understanding of community medicine, but also functions as a motivating force by exposing them to the expectations of community people.
**Takamura et al. (2015)** [48]	Final year/4 months/ 281 people/km^2^	To clarify the challenges of the application of an integrated longitudinal clerkship in a rural community hospital	One of the most important outcomes of the LIC was an enhanced understanding of the community-based practice and the community itself, especially in a rural setting.
**Saiki et al. (2016)** [49]	First year/5 days/ 1963 people/km^2^	To explore how a longitudinal interaction with citizens can develop medical students’ communication skills and understanding of themselves as social entities.	Medical students developed an understanding of citizen-centered communication and of the human relationship and capacity to expand one’s world via social existence.
**Ohta et al****.****(2018)** [50]	Resident/1 week/ 43.1 people/km^2^	To clarify learning content in CBME on a rural island	A strong connection among the islanders, islander-centered care, and the differences between rural and hospital medicine were the main aspects remarked upon by those who had experienced deep relationships with the islanders.
**Yamada et al****.****(2018)** [51]	First year/5 days/ 26 people/km^2^	To investigate medical students’ changes in perception regarding their living	Medical students changed their perception regarding rural medicine and their learning approach to medical science.
**Ohta et al****.****(2019)** [52]	Clinical and final years/10 days/ 64.4 people/km^2^	To investigate the changes in perceptions of participants who completed a two-week CBME course.	The participants’ ratings regarding community care improved significantly from pre-to post-training. The participants realized the importance of community care and of having respect for individuals’ lifestyles.
**Mixed-Method**			
**Takamura et al. (2017)** [53]	Resident/2 years/ 281.3 people/km^2^	To explore the effects of introducing community members to medical education as active teachers.	The participants scored higher regarding their views on the importance of and their preferences for working with communities. Important themes that emerged from the interns’ interviews were taking responsibility for shared understanding, community-oriented focus, valuing community nurses, and tension from competing demands.
**Ohta et al****.****(2019)** [54]	First and pre-clinical years/3 days/ 126 people/km^2^	To clarify the short-term learning experienced by medical students on rural islands in Japan	By interacting with various islanders, students developed an understanding of the different cultural backgrounds in which rural physicians worked, in addition to establishing their own ability to adapt to each. This experience motivated them to pursue studies on rural medicine upon their return to the mainland.

## Data Availability

All relevant data sets in this study are described in the manuscript.

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
