# Peer review of "The Contribution of Citizens to Community-Based Medical Education in Japan: A Systematic Review"

_ijerph, 2021, doi:10.3390/ijerph18041575_

Round 1

Reviewer 1 Report

The authors designed a systematic review about community-based medical education, that is helpful for supporting healthcare professionals in aging societies.

The proposed approach is interesting but there are some points that the authors have to better discuss.

The authors should be better described the novelties of their approach with respect to existing ones. In particular, the author should discuss limitation and cons that their approach aims to overcome at the end of the Related Works section. Furthermore, the authors should provide more details and discussion about the obtained results. The Discussion section also needs to be improved by analyzing the outcome of evaluation section.

I suggest to analyze also more recent approaches about the examined topics. In particular, I suggest to further investigate community approaches based on multimedia analysis, that has been investigated in these papers:

1) DICO: a graph-db framework for community detection on big scholarly data. IEEE Transactions on Emerging Topics in Computing.

2) Community detection based on game theory. Engineering Applications of Artificial Intelligence, 85, 773-782.

3) Multimedia story creation on social networks. Future Generation Computer Systems, 86, 412-420.

Finally, I suggest to perform a linguistic revision.

Author Response

The authors designed a systematic review about community-based medical education, that is helpful for supporting healthcare professionals in aging societies.

The proposed approach is interesting but there are some points that the authors have to better discuss.

The authors should be better described the novelties of their approach with respect to existing ones. In particular, the author should discuss limitation and cons that their approach aims to overcome at the end of the Related Works section. Furthermore, the authors should provide more details and discussion about the obtained results. The Discussion section also needs to be improved by analyzing the outcome of evaluation section.

Response:

We thank the reviewer for this insightful comment. We agree with the suggestion. We have added a description regarding the suggestions for the following research including setting patient reported outcomes such as quality of life and self-rated health in the limitation. Furthermore, we have added further description regarding the evaluation section in the discussion.

I suggest to analyze also more recent approaches about the examined topics. In particular, I suggest to further investigate community approaches based on multimedia analysis, that has been investigated in these papers:

1) DICO: a graph-db framework for community detection on big scholarly data. IEEE Transactions on Emerging Topics in Computing.

2) Community detection based on game theory. Engineering Applications of Artificial Intelligence, 85, 773-782.

3) Multimedia story creation on social networks. Future Generation Computer Systems, 86, 412-420.

Response:

We thank the reviewer for this insightful comment. We agree with the suggestion. We have revised the discussion to include the suggested article so as to discuss the assessment by using multimedia analysis.

Finally, I suggest to perform a linguistic revision.

Response:

We thank the reviewer for this insightful comment. We agree with the suggestion. We have revised the entire article in regard to English proficiency.

Reviewer 2 Report

Thank you for the opportunity to evaluate the manuscript. The work concerns an important aspect of teaching and motivating medical personnel. The structure of the review is correct. The methodology was described and the material was selected.

I have some comments / recommendations:
- in the current pandemic situation, the CBME methodology is significantly limited. The literature review concerns training and internships from 2004-2019. The authors did not take up the topic of further work in the COVID-19 pandemic state. I propose to add the authors' recommendations in the DISCUSSION section.

- There are 67 items in the literature, but only 18 (27%) refer to articles newer than 2016. I recommend supplementing the literature with the latest items, including:

a) Hatano, Y., et al. , The vanguard of community-based integrated care in Japan: The effect of a rural town on national policy. International journal of integrated care, 2017.17 (2): p. 2.
(community integrated care system in Japan)

b) Klepacka, M., Bakalarski, P., Trust of society towards selected medical professions - doctors, nurses, paramedics. Crit. Care Innov. 2018.1 (2): p. 1-10.
(public trust in medical personnel may determine the effectiveness of cooperation during the therapeutic procedure, and also affect the quality of medical education)

c) Saiki, T., Imafuku, R., Suzuki, Y., and Ban, N., The truth lies somewhere in the middle: swinging between globalization and regionalization of medical education in Japan. Medical Teacher, 2017. 39 (10): p. 1016-1022.
(regionalization of medical education in Japan)

d) Mitura, K., The impact of COVID-19 pandemic on critical care and surgical services availability. Crit. Care Innov. 2020.3 (2): p. 43-50.
(impact of the pandemic on healthcare)

e) Martin, A., Lang, E., Ramsauer, B., Gröning, T., Bedin, G. L., & Frank, J., Continuing medical and student education in dermatology during the coronavirus pandemic – a major challenge. JDDG: Journal der Deutschen Dermatologischen Gesellschaft, 2020.18 (8): p. 835-840.
(medical education during a pandemic)

Author Response

Thank you for the opportunity to evaluate the manuscript. The work concerns an important aspect of teaching and motivating medical personnel. The structure of the review is correct. The methodology was described and the material was selected.

I have some comments / recommendations:
- in the current pandemic situation, the CBME methodology is significantly limited. The literature review concerns training and internships from 2004-2019. The authors did not take up the topic of further work in the COVID-19 pandemic state. I propose to add the authors' recommendations in the DISCUSSION section.

Response:

We thank the reviewer for this insightful comment. We agree with the suggestion. We have revised the discussion and added content regarding COVID-19.

- There are 67 items in the literature, but only 18 (27%) refer to articles newer than 2016. I recommend supplementing the literature with the latest items, including:

  1. a) Hatano, Y., et al. , The vanguard of community-based integrated care in Japan: The effect of a rural town on national policy. International journal of integrated care, 2017.17 (2): p. 2.
    (community integrated care system in Japan)
  2. b) Klepacka, M., Bakalarski, P., Trust of society towards selected medical professions - doctors, nurses, paramedics. Crit. Care Innov. 2018.1 (2): p. 1-10.
    (public trust in medical personnel may determine the effectiveness of cooperation during the therapeutic procedure, and also affect the quality of medical education)
  3. c) Saiki, T., Imafuku, R., Suzuki, Y., and Ban, N., The truth lies somewhere in the middle: swinging between globalization and regionalization of medical education in Japan. Medical Teacher, 2017. 39 (10): p. 1016-1022.
    (regionalization of medical education in Japan)
  4. d) Mitura, K., The impact of COVID-19 pandemic on critical care and surgical services availability. Crit. Care Innov. 2020.3 (2): p. 43-50.
    (impact of the pandemic on healthcare)
  5. e) Martin, A., Lang, E., Ramsauer, B., Gröning, T., Bedin, G. L., & Frank, J., Continuing medical and student education in dermatology during the coronavirus pandemic – a major challenge. JDDG: Journal der Deutschen Dermatologischen Gesellschaft, 2020.18 (8): p. 835-840.
    (medical education during a pandemic)

Response:

We thank the reviewer for this insightful comment. We agree with the suggestion. We have added the suggested references to our manuscript.

Reviewer 3 Report

This systematic review has been focused on papers published regarding the Japanese experience on community-based medical education and specifically involving citizens from communities. The topic is interesting and new. However, it is too specific and does not apply to all countries, only those with an ageing population and a high level of development. This point must be described by the authors and implications could be presented in the discussion section.

a critical point is what citizens involvement means, how has been considered and what possibilities of generalizing these results to other societies the authors believe exist.

Data synthesis must be described.

It appears from the results that there was no repetition of references. Was this the case?

What does the training of doctors have to focus on? Are there cultural factors in Japan that are very specific to Japanese society? How can these results be useful to other countries? What elements are critical to the success of these approaches? All these questions could be replied.

Author Response

This systematic review has been focused on papers published regarding the Japanese experience on community-based medical education and specifically involving citizens from communities. The topic is interesting and new. However, it is too specific and does not apply to all countries, only those with an ageing population and a high level of development. This point must be described by the authors and implications could be presented in the discussion section.

Response:

We thank the reviewer for this insightful comment. We agree with the suggestion. We have revised our discussion of the limitations of our study by adding implications of our study in regard to other countries’ conditions.

a critical point is what citizens involvement means, how has been considered and what possibilities of generalizing these results to other societies the authors believe exist.

Response:

We thank the reviewer for this insightful comment. We agree with the suggestion. We have revised the introduction and discussion sections to show what citizen involvement means as well as the applicability of the results to other societies.

Data synthesis must be described.

Response:

We thank the reviewer for this insightful comment. We agree with the suggestion. We have revised the result and limitation portions to clearly describe the synthesis and difficulty of metanalysis.

It appears from the results that there was no repetition of references. Was this the case?

Response:

We thank the reviewer for this insightful comment. We agree with the suggestion. We have revised the entire list of references.

What does the training of doctors have to focus on? Are there cultural factors in Japan that are very specific to Japanese society? How can these results be useful to other countries? What elements are critical to the success of these approaches? All these questions could be replied.

Response:

We thank the reviewer for this insightful comment. We agree with the suggestion. We have revised the entire manuscript regarding the focus of physicians’ education, the aging situation in Japan including the applicability to other contexts, and the importance of citizen involvement in CBME.

Reviewer 4 Report

Thanks for the opportunity to review this paper.  Community based medical education is an important component of health professional training internationally.  This paper undertakes a systematic review of the contribution of citizens to CBME in Japan.

The focus of this paper is on the older person - the reasons for this have been outlined in the introduction.  The authors may consider including some statistics about the Japanese population to help the reader understand the context.

The inclusion of information about Japanese literature search systems is very useful for readers who are unfamiliar to search engines that are not English.  Could the authors provide reasons for not including search engines such as CINAHL, Web of Science, Scopus and Embase.  I would suggest that the authors ensure that all possible articles are found.

Line 127:  The authors should consider the inclusion focused on outlining the Kirkpatrick Model - This could allow a broader readership and understanding of the article.

Table 1 - Population - residents - could the authors clarify if this refers to medical residents or all health professional residents.

Line 143-144 - could the authors include the approval number (if available)

Table 2 - I would suggest that the authors look at the formatting of this table and clearly relate the number to the 21 articles being reviewed.

Table 6 - could the authors consider changing the orientation of this table to landscape?  Categorization of similar articles should also be considered.

While the discussion of the results is good, I would suggest that the authors consider the inclusion a section that specifically outlines the limitations of the study.

Could the authors discuss their results in the context of international medical education literature?

Author Response

Thanks for the opportunity to review this paper.  Community based medical education is an important component of health professional training internationally.  This paper undertakes a systematic review of the contribution of citizens to CBME in Japan.

The focus of this paper is on the older person - the reasons for this have been outlined in the introduction.  The authors may consider including some statistics about the Japanese population to help the reader understand the context.

Response:

We thank the reviewer for this insightful comment. We agree with the suggestion. We have added the rate of people over 65 years old in Japan.

The inclusion of information about Japanese literature search systems is very useful for readers who are unfamiliar to search engines that are not English.  Could the authors provide reasons for not including search engines such as CINAHL, Web of Science, Scopus and Embase.  I would suggest that the authors ensure that all possible articles are found.

Response:

We thank the reviewer for this insightful comment. We agree with the suggestion. We have searched the suggested search engines of CINAHL, Web of Science, Scopus, and Embase from April 1990 up to August 2020, And we have thus revised the search strategy section.

Line 127:  The authors should consider the inclusion focused on outlining the Kirkpatrick Model - This could allow a broader readership and understanding of the article.

Response:

We thank the reviewer for this insightful comment. We agree with the suggestion. We have added an explanation of the Kirkpatrick model in the method section and inclusion criteria.

Table 1 - Population - residents - could the authors clarify if this refers to medical residents or all health professional residents.

Response:

We thank the reviewer for this insightful comment. We agree with the suggestion. We have revised the phrase “residents” to “medical residents.”

Line 143-144 - could the authors include the approval number (if available)

Response:

We thank the reviewer for this insightful comment. We agree with the suggestion. We have added the approval number.

Table 2 - I would suggest that the authors look at the formatting of this table and clearly relate the number to the 21 articles being reviewed.

Response:

We thank the reviewer for this insightful comment. We agree with the suggestion. Some research contained a variety of participants ranging across several grades, from first to fourth year, etc. We have added a description of the calculation approach in the method section.

Table 6 - could the authors consider changing the orientation of this table to landscape?  Categorization of similar articles should also be considered.

Response:

We thank the reviewer for this insightful comment. We agree with the suggestion. We have revised Table 6 and categorized the selected articles based on research methodology.

While the discussion of the results is good, I would suggest that the authors consider the inclusion a section that specifically outlines the limitations of the study.

Response:

We thank the reviewer for this insightful comment. We agree with the suggestion. We have added a limitation section outlining the limitations of our research.

Could the authors discuss their results in the context of international medical education literature?

Response:

We thank the reviewer for this insightful comment. We agree with the suggestion. We have revised the discussion section to include international contexts in respect to the difference of such contexts.

Round 2

Reviewer 1 Report

I think that the authors have addressed all my concerns

Reviewer 3 Report

Authors have included details and changes in the manuscripit. All them are right.